# The Close Possibility of Time Travel

## Nikk Effingham [1,2] 

1   Department of Philosophy, University of Birmingham, Birmingham B15 2TT, UK; nikk.effingham@gmail.com
2   Department of Philosophy, University of Johannesburg, P.O. Box 254, Johannesburg 2006, South Africa

**Abstract:** This article discusses the possibility of some outlandish tropes from time travel fiction, such as people reversing in age as they time travel or the universe being destroyed because a time traveler kills their ancestor. First, I discuss what type of possibility we might have in mind, detailing 'close possibility' as one such candidate. Secondly, I argue that—with only little exception—these more outlandish tropes fail to be closely possible. Thirdly, I discuss whether these outlandish tropes may nevertheless be more broadly possible (e.g., metaphysically or logically possible), arguing that whether they are or not depends upon your favored metaphysics of the laws of nature.

**Keywords:** time travel; laws of nature; modality; philosophy of fiction

## 1. Introduction

David Lewis was interested not only in time travel's logical possibility but also 'with the [possibility of the] sort of time travel that is recounted in science fiction' [1] (p. 145). I believe that Lewis saw these as two distinct enterprises, my evidence being him repeatedly making clear his specific interest in the stories themselves [2] (pp. 441, 454). Certainly, other philosophers have explicitly investigated the possibility of specific time travel fictions, as well as the possibility of tropes from those fictions [3–7]. This paper is in that same vein, focusing on the possibility of various tropes common to time travel fiction.

There are three parts to this paper. Part one (Sections 2 and 3) examines the different types of possibility, introducing 'close possibility'. Part two (Sections 4–7) introduces tropes drawn from fiction and discusses whether they are closely possible or not. I end in part three (Section 8) by considering the broader possibility of those tropes, arguing that the answer depends upon one's views about the metaphysics of the laws of nature.

## 2. Fictional Tropes & Worlds
### 2.1. Fictional Tropes

Whilst 'trope' is sometimes used in metaphysics to refer to a type of property, in this paper, I exclusively use it to refer to 'literary tropes', i.e., elements common to different narratives. For instance, it is a common trope in fantasy that a hero is a 'chosen one', and a common trope in romcoms is that initially, antagonistic people fall deeply in love. This paper is specifically interested in the possibility of the tropes themselves. I assume that a fictional trope is possible iff some possible fiction featuring that trope is itself possible.

### 2.2. Fictional Worlds

I also help myself to 'world talk', talking of 'fictional worlds'. Roughly speaking, (where 'φ' is a variable for a proposition, $w$ for a world, and F for a fiction):

World $w$ is the fictional world depicting fiction F = $_{df}$ $\forall$ φ (φ is true at $w$ iff φ is true according to F)

Three notes are in order.

Note one: I am agnostic regarding which theory of fictional truth is correct [8], assuming only that there are implicit fictional truths [9] (p. 52n1). Implicit fictional truths are those that are not explicitly stated in the fiction. Lewis's example [10] (p. 41) is of Holmes

living closer to Paddington Station than Waterloo Station—that is not explicit in the text but follows from the actual fact that Baker Street is closer to Paddington than Waterloo. Similarly, we can include amongst the implicit truths those subtle truths that only become clear upon close reading of the text (e.g., Deckard being a replicant in *Bladerunner*).

Note two: Some implicit fictional truths are explanatory truths. For example, according to *Harry Potter*, certain things are explicitly true (e.g., that Cedric Diggory drops dead when Peter Pettigrew intones the words 'avada kedavra'), but there are also implicit explanatory facts (e.g., Diggory drops dead *because* those words are intoned). Those implicit explanatory facts distinguish the fictional world of *Harry Potter* from, say, a physically possible world at which exceedingly unlikely quantum fluctuations coincidentally cause people to drop dead when, by happenstance, those around them intone certain phrases. When I come to discuss the tropes below, I assume that the tropes are implicitly explained by time travel having taken place. For instance, when I discuss the deaging trope, I will assume that agents deage *because* they travelled back in time.

Note three: Fictions are incomplete. According to (almost) every fiction, some propositions are neither true nor false, e.g., it is neither true nor false that Harry Potter wears size nine shoes. Since every possible world is complete, it seems that strictly speaking, there are no fictional worlds; at best, there is a *set of* worlds, $s$, depicting a fiction F (where $w \in s$ iff everything true/false according to F is true/false according to $w$). Since incompleteness is tangential to the issues at hand, and because meticulous attention to this technicality would serve only to obstruct, I simply ignore incompleteness and pretend that precisely one possible world bears out any given fiction's narrative (which, in any case, is a meritable theory in its own right [11]).

## 3. Close Possibility

When we ask whether fictions/tropes are possible, we can have in mind different types of possibility. Physical possibility is only one such candidate (e.g., Klimov's *Come and See* is physically possible whilst Lucas's *Star Wars* is not); moreover, in the context of science fiction, it is unlikely to be of much interest. Certainly, when it comes to time travel, broader modalities are normally our focus, e.g., logical possibility [1,5,12–17], metaphysical possibility [3,18–23], conceptual possibility [24], or absolute possibility [25,26].

However, this does not exhaust the relevant types of possibility. Metaphysicians are sometimes interested in a modality broader than physical possibility but narrower than logical/metaphysical/absolute possibility—see, for instance, Lewis's discussion of 'inner spheres' of worlds [27] (p. 475) (see also [28,29]). This paper likewise focuses on such a modality, which I call 'close possibility':

$\varphi$ is closely possible = $_{df}$ $\varphi$ is true at some world with laws of nature similar to the actual laws of nature.

See Figure 1 [1].

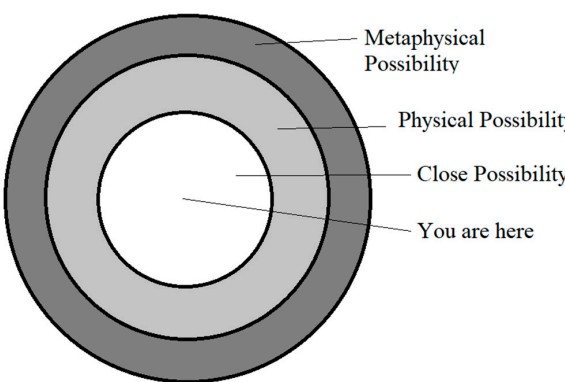

**Figure 1.** Varieties of modalities.

However, just because one can define a type of possibility does not mean that it is, thereby, worthy of investigation. That said, this section considers three reasons for thinking that close possibility is worthy of in-depth consideration.

### 3.1. Philosophy of Science Fiction: Analysing 'Hard Science Fiction'

One question in the philosophy of science fiction is what it takes for something to be a work of science fiction [30,31]. Presumably, then, we may further ask what it takes to belong to a specific sub-genre of science fiction, e.g., space opera, cyberpunk, New Wave, etc. One sub-genre is 'hard science fiction'—see *inter alia*, the works of Stephen Baxter, Arthur C. Clarke, Kim Stanley Robinson, and Annalee Newitz.

A (flawed) proposal is that a work is hard science fiction iff it is both science fiction and physically possible. This analysis would have two problems.

First: An author may aim for their fiction to be physically possible only for later discoveries to reveal that it is not, in which case it would be churlish to recategorise the fictional work. For example, some consider [32,33] Crichton's *Jurassic Park* [34,35] to be hard science fiction, even though paleontological research has revealed that dinosaurs had feathers, which conflicts with Crichton's depiction of featherless dinosaurs. Presumably, though, this contemporary revelation is irrelevant to the fictional work's categorisation.

Second: Some hard science fiction is *intentionally* physically impossible. In Egan's Orthogonal trilogy [36–38], the fundamental laws of physics are the same as ours except that spacetime intervals (calculated one way at our world: $\sqrt{[x^2 + y^2 + z^2 - (ct)^2]}$) are calculated differently (namely: $\sqrt{[x^2 + y^2 + z^2 + (ct)^2]}$). During the course of the trilogy, Egan meticulously explains how this tweak results in radical macroscopic differences, e.g., plants emitting light, objects that travel at infinite velocity, and perpetual motion. Even though it is explicitly and knowingly physically impossible, the fiction is clearly hard science fiction (although, see below).

Close possibility offers a solution to both problems. Consider:

ANALYSIS Fiction F is hard science fiction = $_{df}$ F is closely possible, and F is science fiction.

ANALYSIS solves the first problem. The biological laws of *Jurassic Park* may not be the same as our world's, but they are clearly similar. ANALYSIS equally solves the second problem. The laws true at Egan's fictional world are almost exactly similar to our own; thus, it is closely possible and Egan correctly comes out as a hard science fiction author.

There is a wrinkle: close possibility is analysed in terms of similarity; similarity is context-dependent; thus, close possibility (and, by extension, 'hard science fiction') is likewise context-dependent. But this is a feature, not a bug. Sometimes, people argue about whether something is hard science fiction or not. For instance, someone might argue that Egan is *not* writing hard science fiction and, in writing a book set in a universe with altered laws, is instead writing speculative fiction. Or consider Iain M. Banks and his Culture novels, such as *Consider Phlebas* [39]. Some people think Banks is writing hard science fiction [40–42], but we might disagree (because, say, in *Consider Phlebas*, there are anti-gravity belts that ignore the force produced by gravitational mass but not the forces applied by acceleration, which makes no sense given actual physics). Such disputes seem somehow 'lightweight' rather than a genuine and serious debate. One way for debates to be lightweight is for them to result from disputants being in different contexts; for instance, a five-year-old who says I am tall and a professional basketball player who says I am not are not having a genuine or serious disagreement because they are speaking in different contexts. So, by endorsing ANALYSIS—and by making context core to the definition of 'hard science fiction'—we can allow that these disputes are similarly lightweight. When someone says that the Orthogonal trilogy is *not* hard science fiction, that is because they are taking the laws to be 'appropriately similar' iff they result in similar macroscopic results to those observed at the actual world; since they do not, then *in that context* Egan is not a hard science fiction author. When I say otherwise, I instead treat the laws as being 'appropriately similar' iff the fundamental laws are roughly the same, in which case Egan's world *is*

closely possible. Another example: When it comes to Banks, precisely the reverse is the case. In a context where laws are similar iff they produce similar macroscopic results, Banks is a hard science fiction author; in a context where the fundamental laws must be roughly the same, he is not. Context shift explains these disagreements about what is or is not hard science fiction and, likewise, captures the lightweight nature of the dispute.

Similarly, it still allows for some 'heavyweight' disagreements. If someone says that *Sense and Sensibility* is hard science fiction, then, given ANALYSIS, they are wrong regardless of context because of ANALYSIS's second conjunct. Moreover, that wrongness is not 'lightweight' like the previous disputes—if I correct you about *Sense and Sensibility*, I am being genuinely informative in a way that I am not being when I try and 'inform' you that Banks is not a hard science fiction author.

### 3.2. Aesthetics: The Value of Close Possibility to Writers

For some writers, the close possibility of a fiction ties into its aesthetic value. As evidence, consider these three things:

*Exhibit one:* The existence of hard science fiction is itself evidence. It is a popular sub-genre, demonstrating that close possibility is prized by authors and audiences alike.

*Exhibit two:* Time travel stories are often judged according to whether their laws are chaotic/arbitrary or consistent/well-worked out (i.e., whether they are similar to the actual laws of nature). Damning reviews of films like *Timecop 2: The Berlin Decision* [43] and *A Needle in the Haystack* [44] make clear that if the 'fictional laws' of time travel are arbitrary, this is thought to be an aesthetic flaw. Similarly, we can take as evidence those authors who stress the importance to narrative tension of ensuring consistent and non-chaotic laws of time travel [45–47].

*Exhibit three:* I have consulted on multiple occasions for film and television. In each case the brief has been the same: Were time travel real, how would it work? Given the standard Lewis-Stalnaker analysis of counterfactuals, this amounts to asking what the laws concerning time travel are like at the closest possible world to actuality; this brief is very similar to (though, admittedly, not exactly the same as) asking which tropes of time travel are closely possible versus which are not.

Thus, there is merit in determining the close possibility of a fiction/trope insofar as it helps gauge the aesthetic value of fictions (or, at the very least, it helps those authors *who believe* that it helps gauge the aesthetic value of fictions).

Note that I do not claim that failing to be closely possible is necessarily of aesthetic disvalue. Whilst it may be of value in some cases, close possibility seems clearly irrelevant to, e.g., light-hearted comedies [48–52], children's fiction [53,54], and magical realist fiction [55–58].

### 3.3. Metaphysics: The Theoretical Sandbox

Considering close possibility can deepen our understanding of the actual world. For example, we often imagine Newtonian physics is true or that the Earth is flat (or both [59]!) as part of our method of learning useful things about an actual situation. Or consider what Greg Egan says of his own trilogy:

> "*What value is there in studying a universe ruled by laws different than our own? [ . . . ] We can deepen our understanding of many things—whether it's a human society, a planet's geology, the chemistry of life, or the most basic laws of physics—by striving to imagine how they might be different.*" [60]

Such sentiments justify the general philosophical project of investigating the close possibility of fictions. One may doubt, though, whether they justify this paper's project. Because the time travel tropes examined in Sections 4–6 turn out to *not* be closely possible, one may question the merit of my particular investigation. My response comes in Section 8: Whilst the investigation may not tell us about the *physics* of the actual world, consideration of the tropes nevertheless deepens our understanding of *metaphysics*.

#### 4. Deaging/Aging

Each of the next three sections introduces a different time travel trope, arguing that it is not closely possible (with a slight exception made in Section 6.2). I start with the trope of time travel affecting one's biological age.

*4.1. Examples of the Trope*

In some fictions, one 'deages' as one travels back in time, getting younger and younger the further back one goes. This trope appears in one of the first time travel fictions, Gaspar's *El Anacronópete* [61]. It has appeared in other fictions since then [62,63]. There are also variations of this trope, such as time travelers aging rather than deaging [64–66] or randomly aging/deaging [67,68]. A further variation is to have a device (or power) connected to time travel allowing someone to cause objects/people to age/deage. For instance, the titular character in *Lost in Space*'s 'The Time Merchant' has the ability to return someone to their youth [69]. Other examples are commonplace [70–74] [75] (pp. 43–53) [76] (p. 81) [77] (p. 76) (And a sub-variation on this trope is for the device/power to cause things to evolve/devolve rather than age/deage [78,79] [80] (pp. 32–33) [2].)

*4.2. Deaging & Close Possibility*

The remainder of this section argues that this trope is *not* closely possible, using Trouwborst's *Captain Nova* [63] as an example. In *Captain Nova*, time travelers revert to the age they were originally at the point of time at which they arrive in the past. For instance, Nova is a thirty-seven-year-old woman who travels back in time twenty-five years, arriving in the past to discover that she is now a twelve-year-old child.

I will discuss two problems for the close possibility of this fiction, one which is less problematic and one which is more problematic.

The first, more minor, problem concerns Nova's brain. Time travel causes Nova's body to regress, but her mind is unaffected. Assuming physicalism, either her brain remains unaltered, or it alters in a very specific way that keeps her mental state unchanged. Both options are arbitrary. The former is arbitrary because, whilst the rest of her body (her kidneys, liver, heart, etc.) all change in size (and undergo other changes regarding the age of the organs), a single part of her—her brain—remains unaffected for no apparent reason. The latter is arbitrary because it is arbitrary for Nova's brain to change and yet nevertheless retain its earlier functionality whilst her other organs do not do that—for instance, presumably, her age-regressed heart now pumps as much blood as an average twelve-year-old's would, her bladder now holds less than that which her adult bladder previously held, her eyesight is not the presbyopic sight of an adult, etc. To avoid this arbitrariness, we should instead deny physicalism. At the world of *Captain Nova*, substance dualism is true [3].

Were substance dualism true at the fictional world, extra psychophysical laws must also be true. This alone is a problem for the close possibility of the fictional world, though I shall not press this point too hard. Close possibility is context-sensitive, so there are competing, equally reasonable interpretations of close possibility. I will charitably assume that, under some of them, the relevant psychophysical laws will count as being similar to the laws of our physicalist actual world. Thus, a commitment to substance dualism alone does not mean that a world fails to be closely possible.

Indeed, I should say something about what I will take 'appropriately similar laws' to mean in this paper. Charity dictates a liberal approach to appropriate similarity. Were I too conservative, people with more liberal views on close possibility would have no reason to accept the conclusions of this paper. A liberal understanding of similarity would be to say that laws are similar iff they have the same *metaphysical character*. 'Metaphysical character' is slightly nebulous, but I can flesh out the idea well enough for the purposes of this article. Vis-à-vis metaphysics, laws of nature have different characteristics. They might be ontologically reducible, such that higher-level laws are reducible to the laws of nature governing more fundamental things. They might be ontologically egalitarian, applying

to all objects equally. They might be invariant across space and time. And so on. These very general, metaphysically relevant characteristics are shared by the actual laws. In the context I am aiming for, having such features is what makes them appropriately similar. And, presumably, any reasonable interpretation of 'appropriate similarity' will at least have it that such characteristics are shared; thus, by fixing appropriate similarity at the level of sharing such general characteristics, we have a very liberal—and, therefore, very charitable—assumption of what close possibility amounts to. (And it is a standard which I believe a substance dualist world probably meets—at least if we assume that Cartesian souls are mereologically simple fundamentalia.)

That said, move to the second problem. The actual fundamental laws of nature do not feature *sortal terms*; they may dictate what happens to an object that is charged, or an object that spins up, or an object with such-and-such a mass, but they do not dictate what happens to different *types* of objects. Similarly, those laws are ontologically egalitarian—the sort of thing an object is makes no difference to how it interacts with the fundamental laws of physics.

That said, consider the following law:

LAW$_1$ Anything that travels back in time $n$ years regresses in age $n$ years.

LAW$_1$ is ontologically egalitarian and mentions no sortal terms; it is of the same character as the actual laws. But LAW$_1$ is not the law that is true at the fictional world of *Captain Nova*. Nova regresses twenty-five years, but she is the *only* object that regresses twenty-five years. In a human body, blood cells live for up to one hundred and twenty days, liver cells for up to eighteen months, cells in the heart for forty years, fat cells for eight years, etc. Given LAW$_1$, they would all have to regress twenty-five years. Rather than a twelve-year-old girl arriving in the past, what would arrive would be a pool of inchoate biological matter. Similarly, the non-biological objects fail to regress. Nova travels back in time in a spaceship-style time machine that is unaffected by temporal regression. Since the spaceship did not even *exist* twenty-five years ago, it should have ceased to be if LAW$_1$ were true.

One move would be to say that LAW$_1$ applies only to the fundamental objects of physics, e.g., the sub-atomic particles making up Nova and the spaceship. That the laws of nature have such restricted domains is probably true of the actual laws of nature. For instance, Newton's laws of motion are a law of nature, but whilst they apply to sub-atomic particles (and some composite objects like lumps of clay), they (arguably) do not apply to constituted objects like statues [97] (pp. 112–113) [98]. Similarly, there is a ban on things travelling faster than the speed of light, but we might presume that this does not apply to certain things, e.g., shadows and gerrymandered objects [99] (pp. 21–22). But even if we allow this and have it that LAW$_1$ is restricted to sub-atomic particles, it would not explain the fictional narrative. When Nova goes back in time, her sub-atomic particles would all deage twenty-five years. Because the intrinsic properties of her quarks and superstrings are (let us assume) invariant over time, she herself would suffer *no* change and would not regress. (And, even if you ditched the assumption of invariance, it would nevertheless be difficult to see how those sub-atomic changes would result in Nova turning into a twelve-year-old whilst her spaceship remained unaffected!)

A different approach is to tweak the law:

LAW$_2$ Any organism that travels back in time $n$ years regresses in age $n$ years.

Given LAW$_2$, Nova regresses in age, but her organelles, cells, organs, etc. do not (and nor does the spaceship). The problem with LAW$_2$ is that it is ontologically inegalitarian (treating certain composites, i.e., organisms, differently from other composites, e.g., organs and spaceships) and features sortal terms (i.e., '__ is an organism'). This is not of the metaphysical character of the actual laws, so LAW$_2$ fails to be closely possible.

Consider a response: I have been too harsh in saying that the laws of nature are egalitarian in this fashion. Whilst the *fundamental* laws of nature may not mention sortal

terms, and the fundamental laws apply to all objects equally, the laws of nature in general do not have this prohibition. Biological laws (e.g., Mendel's law of genetic inheritance, which concerns parents and children), ecogeographical laws (e.g., Allen's rule, which concerns species and animals), and chemical laws (e.g., Dalton's law of multiple proportions, which concerns elements and compounds) are all inegalitarian in specifically concerning only certain things; such laws clearly feature sortal terms (e.g., 'parent', 'animal', and 'element'). However, these non-fundamental laws are all *reducible* to more fundamental laws. As we consider reducibility, we get to the real problem with deaging laws: they are not reducible! To see why deaging laws are not reducible, let us take $LAW_1$ as an example and set aside the worries just discussed in order to focus solely on the issue of its reducibility. To show that $LAW_1$ is not reducible, I will consider the ways in which $LAW_1$ could be thought to be reducible and then explain why those ways do not work.

Imagine that $LAW_1$ was reducible because the deaging process was the result of both speeding up and reversing existing natural processes. I admit that such a law would be closely possible since it is both closely possible for processes to proceed at a different rate (for it happens given relativity!) and it is closely possible for processes to be time reversible (for it happens given thermodynamics!). Were Nova's biological processes both reversed and sped up, that would explain her regressing back to a twelve-year-old child. Similar thoughts apply in other cases. For instance, when Lister is zapped by a time gauntlet in *Red Dwarf*'s 'Inquisitor' [71], he ages to become a decrepit old man; we might see that as a case of his natural aging process having been radically sped up.

But 'sped-up' processes cannot explain what is going on in these fictions. Sped-up processes require an appropriate interaction with the surrounding environment. Consider Lister. When sped up, he would require vast quantities of food to digest and water to absorb; otherwise, his sped-up body would do what any body does when it goes decades without food and water—it would die and rot away. Similar thoughts apply to Nova, although because she is going in reverse, her time machine would need to launch into the past, carrying twenty-five years of feces and urine for her to "undigest" and "unabsorb" (and, upon arrival, she would have regurgitated twenty thousand meals into the cockpit). Clearly, then, the deaging/aging trope does not involve toying with natural processes in this way [4].

Consider an alternative attempt to make $LAW_1$ reducible. Assume that the world is deterministic. Given determinism, for any object, there is a predictable arrangement that its sub-atomic parts will have at any given point in time. Given that the laws are also reversible, it further follows that there is a specific arrangement that its sub-atomic parts *did* have at any given point in time. Still ignoring the issue of ontological egalitarianism, consider the following law and ask whether it is closely possible:

$LAW_3$ If $x$ time travels back $n$ years, the sub-atomic particles composing $x$ take the position that they had $n$ years ago.

The problem with $LAW_3$ is that parthood is temporary. The sub-atomic parts that Nova had when she was twelve are different from those she had at thirty-seven. If, when time travelling, Nova's parts came to have the arrangement that they had when she was twelve, they would assume some gerrymandered arrangement scattered across the globe. Time travel would kill Nova, not deage her.

An anonymous referee suggested a tweaked alternative to $LAW_3$:

$LAW_4$ If $x$ time travels back $n$ years, then, where the parts of $x$ were previously arranged *F*-wise $n$ years ago, the (personally current) parts of $x$ rearrange themselves to be *F*-wise.

There are two problems with $LAW_4$, one minor and one major.

The minor problem is that, at thirty-seven, Nova had more parts than her twelve-year-old self did, for thirty-seven-year-olds are larger and have more mass than children. When Nova time travels, $LAW_4$ has it that her parts assume an arrangement qualitatively identical to that of her parts from twenty-five years ago; thus, there will be leftover parts.

This means that in *Captain Nova,* we should see leftover biomass, the shedding of flesh, liters of spilled blood, etc. Since this is not what we see on-screen, presumably, LAW$_4$ is not true according to *Captain Nova*.[5] (Similarly, when someone ages via time travel, they can *acquire* parts—where, then, would those parts come from?)

The more major issue concerns what happens when one time travels in the futurewards direction. In *Captain Nova*, when Nova returns to the future, she ages twenty-five years. The corresponding law would be:

LAW$_{4'}$ If $x$ time travels forwards $n$ years, then, where the parts of $x$ will be arranged $F$-wise in $n$ years, the (personally current) parts of $x$ rearrange themselves to be $F$-wise.

LAW$_{4'}$ is problematic. LAW$_4$ only works because, $n$ years ago, Nova's previous self existed, and so there is a fact about what arrangement Nova's parts had at that point. But the same is not true when it comes to her returning to the future. Having left when she time travelled, there is no fact about where her parts are arranged upon her return. Another way of putting the same point: whatever arrangement Nova's parts take upon her return—whether they are arranged twelve-year-old-wise, thirty-seven-year-old-wise, or octopus-wise—then there will be no breach of LAW$_{4'}$ (for, since there is no duplicate of her at that time, whatever arrangement she has is the arrangement she is determined to have!). Thus, problematically, LAW$_{4'}$ does not entail that Nova 'reages' back to being thirty-seven.

This discussion about LAW$_3$/LAW$_4$/LAW$_{4'}$ should signal that to salvage deaging laws, we need to move from laws dealing with *descriptive* facts about what Nova's parts were (or will be) like to (non-moral) *normative* facts about what Nova's parts *should* be like. On this view, for every age that Nova could have, there is an associated fact—a 'blueprint fact'—detailing the precise arrangement of parts she should have at that time. This then allows us to 'fix' the problems with LAW$_4$/LAW$_{4'}$: when Nova time travels, her parts rearrange themselves in accordance with this blueprint fact.

However, blueprint facts are themselves problematic. They appear to be ungrounded 'metaphysical danglers'. For instance, what makes it the case that, presently, Nova is associated with the blueprint that she is actually associated with rather than some subtly different blueprint that differs only over, say, the precise location of some electrons in her bile duct? Moreover, even if these facts did exist, laws of nature engaging with such facts would seem to be radically different from the laws of the actual world, for the actual laws do *not* engage with such strange and bizarre normative facts.

One might object that blueprint facts are not so weird after all. 'He has the kidneys of a man twice his age' and 'your liver is that of a seventy-year-old' are example sentences that indicate that there are normative facts about what one's kidneys and liver *should* be like. Similarly, we might consider biological functional facts, e.g., my heart *should* pump blood around my body, my eye *should* see things around me, etc. If you accept facts like those, you may think that blueprint facts are of the same ilk.

But there are ways in which blueprint facts are quite unlike these sorts of biological functional facts. Firstly, if blueprint facts were like functional facts, then there would be rampant malfunction. Routinely, organs malfunction, e.g., people have eyes that fail to see, and people have hearts that fail to pump (although not for very long) [102] (p. 2504). Similarly, if—somehow, someway—my body had a function requiring my atoms to be in a very specific arrangement, then, routinely, my atoms would *not* be in that very specific arrangement. Since the arrangement of my atoms is altered by my every action (and, due to the effects of gravity, minutely altered by the actions of everyone around me), it would be incalculably unlikely for my atoms to have the precise arrangement dictated by the relevant blueprint fact. Therefore, almost everyone would standardly, routinely, and consistently fail to have the relevant atomic arrangement. Thus, almost everyone would standardly, routinely, and consistently be malfunctioning. And that does not sound right at all—if mostly everyone was malfunctioning yet often flourishing, surviving, and reproducing, would that not make a mockery of the word 'malfunction'?

Secondly, consider the different philosophical theories of function, for example, contributory theories and etiological theories [103].

Contributory theories say that ϕ is a function of *x* iff *x*'s ϕ-ing contributes to something specified by the theory. For instance, it might be that *x*'s ϕ-ing must contribute to the activity of the system it belongs to (for example, my heart has the function of pumping blood because its pumping blood contributes to the activity of the circulatory system to which it belongs [103] (pp. 135–138)) or, alternatively, that *x*'s ϕ-ing must contribute to the survival of the organism to which it belongs [103] (pp. 138–141) (for example, my heart has its function because, if it did not pump blood, I would not survive). Of course, having a specific atomic arrangement would likewise contribute to such things and so qualifies as a function. However, so would many alternative atomic arrangements (for example, would I not be just as liable to survive if a few electrons in my bile duct were not there?). Thus, if any such arrangement qualified as being a function of Nova's body, countless other arrangements would also be a function, with each arrangement being 'on a par' with one another. In that case, contrary to hypothesis, Nova would not have a *unique* blueprint fact. So, contributory theories of function do not bear out blueprint facts.

What of the etiological theories? Etiological theories say that ϕ is a function of *x* iff my ancestor's ϕ-ing led to them surviving and reproducing. Given the etiological theory, it is straightforward that Nova does not have any specific atomic arrangement as a function. Whilst Nova had ancestors who survived to reproduce because of pumping hearts and seeing eyes, no ancestor of Nova had a specific atomic arrangement suitably similar to that of her twelve-year-old self (for, outside of apomictic parthenogenetic organisms such as aphids, creatures are not clones of their ancestors). Therefore, Nova cannot have that function, and there is no appropriate normative fact.

In short, it looks as if the best way to salvage age regression laws requires there to be normative facts interacting with the laws of physics in a weird way. It looks as if the deaging laws fail to be closely possible.

There may be other problems with the close possibility of deaging/aging, but these are enough for us to draw a line under it. As we shall see, similar issues apply to the next two tropes.

## 5. Disaster

### 5.1. Varieties of Disaster

The second time travel trope to consider is that of an alleged paradox causing some terrible disaster. For example, in *Sliders* [104], the protagonists arrive at a universe in which, from their point of view, time intermittently 'jumps back', taking them with it. Once back in the past, they prevent a murder that would have occurred in the future. This results in an (apparent) contradiction, which then causes a tear to open in reality, destroying the universe.

Variations of this trope differ over (i) what types of paradox cause the disaster and (ii) what type of disaster then results. For instance, *Sliders* pairs the paradox of an (apparent) contradiction with a disaster like the universe being destroyed. This is a common pairing [105–108], although poetic license usually saves a handful of people in order for them to witness the end of everything. Likewise, this pairing includes cases where the end of everything is an unrealized threat [109–113] and cases where there are characters billed as experts in time travel who justifiably—yet falsely—believe that such a disaster will occur [114,115] [6].

There are other options when it comes to (i) and (ii). Regarding (i), the disaster-causing paradox is not always an (apparent) contradiction being brought about. The disaster-causing event may instead be a time traveler meeting or touching their earlier self [117–119]) or a time traveler from the future dying at a point prior to when they were born [120]. Regarding (ii), there are three common alternatives to the destruction of the universe. The first is that, like in *Sliders*, a tear opens up in reality, but the tear does not destroy reality, even though it causes other serious problems [120,121]. The second

alternative is that time 'comes apart', with things from different times randomly appearing in the present or things from the present randomly travelling through time [122–124]. The third alternative is that bringing about a paradox summons some sort of creature, which then goes on to wreak havoc [125–127] [128] (p. 285) [7].

*5.2. Logical Explanations for Disasters*

For example purposes, I will continue to use *Sliders*. Call the event of the protagonists of *Sliders* preventing the future murder from occurring '$e_{prevent}$'. Seconds after $e_{prevent}$ occurs, the relevant disaster occurs; call that event '$e_{disaster}$'. As noted in Section 2.1, I assumed that events like $e_{prevent}$ occurring explained events like $e_{disaster}$ occurring. That said, we have a choice: either that explanation is a logical explanation, or it is a physical explanation.

Logical explanations of facts are explanations involving the logical impossibility (or necessity) of a given state of affairs. For example, your inability to make a larger square out of ten smaller squares is explained by ten not being a square number. It is not a physically contingent law of nature that prevents you from making the square, but a 'law of logic'. It is a logical, not a physical, explanation.

Logical explanations already play a prominent role in 'Ludovician' theories of time travel. Ludovicianism is popular throughout fiction—from old stories (e.g., [132–134]), to the not-so-old (e.g., [135–137]), all the way to the new (e.g., [138–140]). Ludovicianism has it that the past cannot be changed: what once was will always have been [1] [18] (pp. 67–73, pp. 116–143) [12,15,141–146]. For instance, if I try to kill Hitler in 1930, I shall fail because, e.g., I slip on a banana peel. Correctly minded Ludovicians say that my slipping on that banana peel is logically explained by the logical impossibility of me successfully assassinating Hitler [147,148].

However, this is *dissimilar* from the case of $e_{prevent}$ causing $e_{disaster}$. For starters, the world of *Sliders* is *not* Ludovician—because the past changes, the world of *Sliders* is 'non-Ludovician'. Non-Ludovicianism is just as popular in fiction as Ludovicianism, likewise appearing in older fictions (e.g., [149–151]), not-so-old fictions (e.g., [152–154]), and newer fictions (e.g., [155–157]). Most such fictions are quiet about what philosophical theory underpins the changeability of the past. Fortunately, metaphysics has not been as mute. Some non-Ludovician philosophers add in extra universes to allow for the past to change [18] (pp. 74–85) [158–160] (which is also the most popular explanation offered in fiction, at least when one is offered at all (e.g., [92,115,118,161,162])). Other non-Ludovicians do without extra universes, instead denying that the earlier-than relation is the converse of the later-than relation [163–165]. However, for reasons I have discussed elsewhere [3] (p. 1318), I will opt for a third alternative: non-Ludovician fictional worlds are best thought of as being 'hypertemporal'.

At a hypertemporal world, there is a second temporal dimension orthogonal to the regular temporal dimension, along which we can change the past [18] (pp. 76–79) [3,4,166–169]. Use upper case *T*s (e.g., $T_1, T_2, T_3 \ldots$) to represent hypertimes and lower case *t*s (e.g., $t_1, t_2, t_3 \ldots$) to represent regular times; use $T_x$-$t_y$ to refer to time $t_y$ at hypertime $T_x$.

Figure 2 depicts the hypertemporal world at which the narrative of *Sliders* is true. There are two hypertimes, $T_1$ and $T_2$. Those hypertimes are at odds as to whether $e_{prevent}$ occurs at $t_1$, whereby the following is true:

$$[\text{At } T_1\text{: at } t_1\text{: } \neg\ e_{prevent} \text{ occurs}] \wedge [\text{At } T_2\text{: at } t_1\text{: } e_{prevent} \text{ occurs}]$$

Notice that this proposition is *not* a contradiction since it is not of the form $\varphi \wedge \neg\varphi$. At the hypertemporal world of *Sliders*, no logically impossible contradictions come about, nor could changing the past ever threaten one. Since there is no threat of a contradiction, this situation is very different from that of Ludovicianism. Thus, there is no reason to think that $e_{disaster}$ is *logically* explained by $e_{prevent}$ occurring.

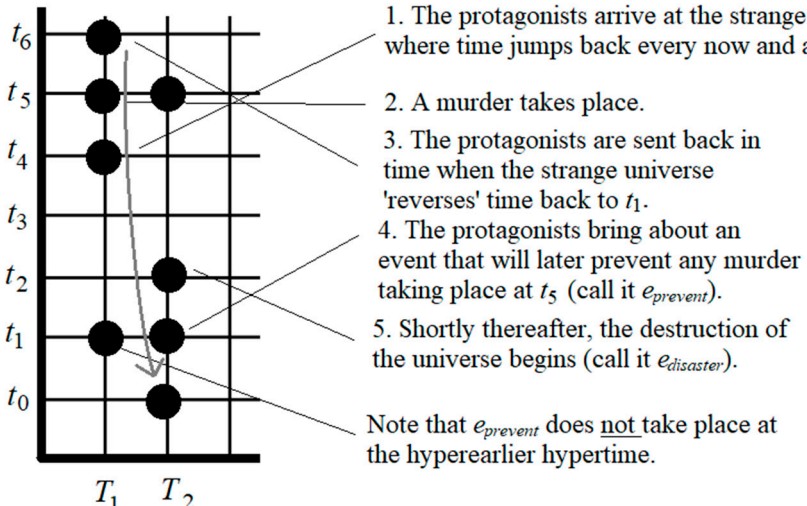

**Figure 2.** The narrative of *Sliders* at a hypertemporal world. The grey arrow indicates time travelers moving from $T_1$-$t_6$ to $T_2$-$t_0$.

*5.3. Physical Explanations for Disasters*

The explanation must, therefore, be a physical explanation. In the same way that the laws of relativity explain why I cannot accelerate faster than the speed of light, some law of nature explains why the universe goes kablooey when one messes around with the past.

Having established that there are no contradictory states of affairs at the fictional world of *Sliders*, it must be some other states of affairs in virtue of which $e_{disaster}$ occurs. Define:

$x$ is a quasi-contradiction (which comes about at $T'$) = $_{df}$ $x$ is the state of affairs of there (i) being two distinct hypertimes, $T$ and $T'$, such that $T$ is the hypertime immediately hyperprior to $T'$ and (ii) there is some time $t$ and some proposition $\varphi$ such that $\langle$At $T$: At $t$: $\varphi \wedge$ At $T'$: At $t$: $\neg\varphi\rangle$ is true [8].

In *Sliders*, there is no contradiction, but (since $e_{prevent}$ does not occur at $T_1$-$t_1$ but does occur at $T_2$-$t_1$) there is a quasi-contradiction. We might then imagine that there is a law saying that when a quasi-contradiction comes about, a disaster occurs. Consider two examples of such a law, which differ over whether the disastrous result is limited only to the hypertime that the quasi-contradiction comes about at or whether it affects all hyperlater hypertimes:

$\mathrm{LAW}_5$ If a quasi-contradiction comes about at $T_m$-$t_n$, then (at $T_m$) everything ceases to be from $t_{n+1}$ onwards (i.e., every hypertime-time $T_m$-$t_{j>n+1}$ contains nothing).

$\mathrm{LAW}_6$ If a quasi-contradiction comes about at $T_m$-$t_n$, then (i) at $T_m$, everything ceases to be from $t_{n+1}$ onwards and (ii) at every hypertime hyperlater than $T_m$, nothing exists at any time.

As I shall now explain, $\mathrm{LAW}_5$ entails that the universe-destroying disaster is not that worrisome. Because the fictions under consideration imply that the disasters that occur *are* worrisome, this means that we should prefer $\mathrm{LAW}_6$ to $\mathrm{LAW}_5$.

$\mathrm{LAW}_5$ entails that the disasters are not that worrisome because, hypereventually, there will be a hypertime where the future is restored. At $T_2$, at times earlier than $t_1$, assume that the world of *Sliders* evolves in a deterministic fashion, mirroring how time evolved at $T_1$.[9] The time travelers arrive at $T_2$-$t_0$, having travelled from $T_1$-$t_6$. At $T_2$-$t_1$, they bring about the relevant quasi-contradiction. Given $\mathrm{LAW}_5$, the universe is then destroyed at $T_2$-$t_2$ but leaves the next hypertime unaffected. Thus, at $T_3$, the universe again deterministically evolves until reaching $t_1$. What happens next depends upon what option we take regarding the existence of the time travelers at $T_3$-$t_0$. There are two options:

*Option One:* The time travelers from $T_1$ also exist at $T_3$, i.e., at $T_3$-$t_0$, out of nowhere, the time travelers appear (just as they did at $T_2$-$t_0$ and as they did *not* at $T_1$-$t_0$). (Argument in favor: whatever laws force hyperlater hypertimes to be qualitatively similar to hyperprevious hypertimes might likewise cause the time travelers' arrival at $T_2$-$t_0$ to be duplicated at $T_3$-$t_0$.)

*Option two:* The time travelers do *not* exist at $T_3$-$t_0$; just as no time travelers arrived at $T_1$-$t_0$, no time travelers arrive at $T_3$-$t_0$. (Argument in favor: The time travelers only arrived at $T_2$-$t_0$ because they came from $T_1$-$t_6$. Since reality is destroyed at $T_2$-$t_2$, no time travelers leave $T_2$-$t_6$; thus, there is no-one to arrive at $T_3$-$t_0$.)

Consider Option one. At $T_3$, the time travelers again prevent the murder, i.e., $e_{prevent}$ takes place at $T_3$-$t_1$. However, since $e_{prevent}$ *did* occur at $T_2$-$t_1$, there is then no quasi-contradiction at $T_3$-$t_1$; hence, no disaster occurs at $T_3$-$t_2$. But, since the universe does not exist at $T_2$-$t_2$ and does exist at $T_3$-$t_2$, we get a new quasi-contradiction; LAW$_5$ then entails that the universe is destroyed at $T_3$-$t_3$. Similar reasoning means the following: at $T_4$, the universe evolves as far as $T_4$-$t_3$ before being destroyed at $T_4$-$t_4$; at $T_5$, the universe evolves as far as $T_5$-$t_4$ before being destroyed at $T_5$-$t_5$; and so on. Eventually, we arrive at a hypertime at which no disaster occurs; the time travelers arrive at $t_0$; and the time travelers prevent a murder at $t_1$ and then... nothing of note happens. The murder is stopped; the time travelers are safe; the universe continues (just as it would at a 'regular' hypertemporal world, i.e., a hypertemporal world where LAW$_5$ is not a law of nature and quasi-contradictions do not bring about universe-ending disasters). Ultimately, time travelers bringing about quasi-contradictions is not a problem because, hypereventually, things will sort themselves out. At the hyperend of the hyperday, blowing up reality just is not all that bad.

Consider Option two, whereby the time travelers do not exist at $T_3$-$t_0$. Since the time travelers existed at $T_2$-$t_0$, we thus get a quasi-contradiction occurring *earlier* than given Option one, i.e., at $T_3$-$t_0$ rather than at $T_3$-$t_1$. Thus, at $T_3$, the universe is destroyed earlier than at $T_2$, ending at $t_1$ instead of $t_2$. However, once we have got over that little snag, then the same thinking as before applies: at every hypertime hypersubsequent to $T_3$, the universe persists for one unit of time more until we get to a hypertime, $T_9$, which is the earliest hypertime at which no disaster occurs at $t_6$. At $T_9$-$t_6$, the time travelers, having arrived at $T_9$-$t_4$ (just as they arrived at $T_1$-$t_4$), will time travel back (to $T_{10}$-$t_0$), cause $e_{prevent}$ (at $T_{10}$-$t_1$) and thus, again, bring about quasi-contradictions that destroy the universe (at $T_{10}$-$t_2$). What we are left with is a reality that 'oscillates': once a universe-destroying disaster occurs, the universe 'regrows' until it reaches some given hypertime and then gets destroyed all over again ad infinitum. That is bad, but not quite as bad as the disaster scenario implied by LAW$_6$—and, presumably, that which is implicitly assumed in the fictions under consideration.

That said, I will assume that LAW$_6$ is the relevant law of nature. Prima facie, LAW$_6$ appears to be closely possible (since it does not seem to be irreducible, does not seem to be ontologically inegalitarian, etc.). But LAW$_6$ nevertheless fails to capture the narrative of *Sliders* because LAW$_6$ entails that *any* time travel destroys reality. When time travel takes place, for some hypertime $T_j$ and times $t_k$ and $t_{l<k}$, time travelers go from $T_j$-$t_k$ to $T_{j+1}$-$t_{l<k}$. Given the non-Ludovician model, they, therefore, change the past (for they were *not* located at $T_j$-$t_{l<k}$ and yet they *are* located at $T_{j+1}$-$t_{l<k}$). Thus, *immediately upon arrival*, the time travelers bring about a quasi-contradiction at $T_{j+1}$-$t_{l<k}$, and LAW$_6$ entails that all of reality is then destroyed. In *Sliders* (and the other referenced fictions), this is not what we see. Some measure of time travel is tolerable in those fictions; reality can put up with some quasi-contradictions more than others; only certain sorts of quasi-contradictions cause disasters. Presumably, the 'disaster causing' quasi-contradictions are those which 'change the course of human history'. Having a cup of tea makes little difference to the future (although see [4] (pp. 7–8)), but life-and-death decisions (e.g., killing or saving someone) make a lot of difference, and, because of this, those sorts of events *do* cause a disaster.

The problem with this approach is that we are then building anthropocentric prejudice into the laws of nature (for squashing a Byzantine spider will not make a huge difference to how World War II plays out, but it makes a huge difference to the spider!). The relevant law would need to be something like:

$LAW_7$ If an anthropocentrically worrisome quasi-contradiction comes about at $T_m\text{-}t_n$, then (i) everything ceases to be from $t_{n+1}$ onwards and (ii) at every hypertime hyperlater than $T_m$, nothing exists at any time.

By making the laws recognise 'anthropocentrically worrisome' as a feature of reality, $LAW_7$ runs into the same sorts of problems involving reducibility as $LAW_2$ faced. Thus, worlds like *Sliders* (and other worlds where laws like $LAW_7$ are true) fail to be closely possible.

## 6. Morphing

Next, I consider the trope whereby time travel causes people/objects to alter, change, or otherwise morph. Sometimes this happens to a time traveler when they are time travelling; other times, it happens to things other than the time traveler themselves. I discuss the two cases separately.

### 6.1. Morphing Time Travelers

The most obvious examples of time travelers 'morphing' are cases where they prevent their birth and cause themselves to vanish. The most famous example is Marty McFly in *Back to the Future* [170], where, having obstructed his parents from getting married, he begins to fade from existence. This trope is commonly repeated [151,171–177].

Variations of this trope are plentiful. The closest variation is where objects, rather than agents, fade away. For example, photographic images in *Back to the Future* [170]; ink fading from paper in its sequel [178]; and blood vanishing from a teddy bear in *Deadpool 2* [179]. Nor are morphings limited to fading and vanishing. A common trope is the 'double memory effect', which is a sub-trope of morphing. It has it that time travelers who change the past suddenly acquire new memories, usually in addition to their old memories [180–183] (or their personality alters but leaves their memories unaffected [184]). A different variation has it that changes to one's past body affect one's future body [112,185] or changes to a past object affect its time-travelling future version. For instance, in *12 Monkeys* [186], the earlier version of a watch is scratched, immediately causing its time-travelling future version to also be scratched.[10] Another example is in *Krypton*, where Superman's cape—brought from the future—morphs to bear Zod's sigil instead of the House of El's, indicating that history has been rewritten to reflect General Zod's dominance [190].

For the purpose of example, I will focus on *The Adam Project* [191]. Adam travels from 2050 to 2022; call that event '$e_{travel}$' and that version of him 'Adam$_{2050}$'. Adam$_{2050}$ then takes his earlier self, 'Adam$_{2022}$', further back to 2018. In 2018, the Adams change the future such that $e_{travel}$ never occurs; call the event causally responsible for that prevention '$e_{halt}$'. A little while after $e_{halt}$ occurs, Adam$_{2050}$ and Adam$_{2022}$ blink out of existence; call that event '$e_{vanish}$'. See Figure 3.

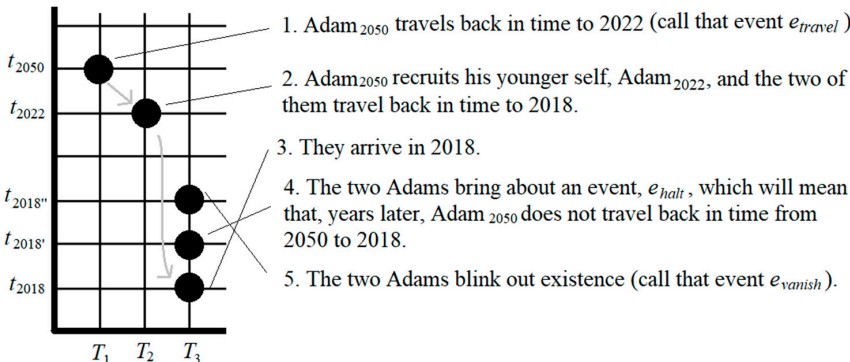

**Figure 3.** The narrative of *The Adam Project* at a hypertemporal world. They grey arrows indicate time travelers going from $T_1$-$t_{2050}$ to $T_2$-$t_{2022}$ and from $T_2$-$t_{2022}$ to $T_3$-$t_{2018}$.

For similar reasons to those given in Section 5.2, $e_{halt}$ occurring does not logically explain $e_{vanish}$ occurring (for nothing about the hypertemporal model itself means that $e_{halt}$ makes a difference to Adam$_{2050}$ and Adam$_{2022}$). The explanation must, therefore, be physical, and there must be a corresponding law of nature, e.g.,:

LAW$_8$ For any event $e$ of a time traveler travelling back in time from $T_j$-$t_k$ to $T_{j+1}$-$t_{l<k}$: if, at a time later than $T_{j+1}$-$t_{l<k}$ (call it $T_{j+1}$-$t_{l<m<k}$), an event occurs that is causally responsible for preventing $e$ from occurring at $T_{j+1}$-$t_k$, then (at $T_{j+1}$-$t_{m<n<k}$) the time traveler ceases to exist.

In *The Adam Project*, $e_{halt}$ prevents $e_{travel}$ from taking place at $T_3$, and so, given LAW$_8$, $e_{vanish}$ then occurs (at $T_3$-$t_{2018''}$).

In its favor, LAW$_8$ is ontologically egalitarian and apparently reducible. This is because Adam's causing of $e_{halt}$ prevents not just himself from travelling into the past but also prevents the atoms composing him from doing likewise; given LAW$_8$, the atoms, therefore, vanish, taking Adam with them. Indeed, with a few tweaks to LAW$_8$, we can explain not just 'blinking out of existence' but also time travelers 'fading away' (which is what more commonly happens in such fictions). In LAW$_8$, some constant must determine the length of time between $e_{halt}$ and Adam's fading; if we imagine that the constant is a function of many different environmental factors or, alternatively, that it is not a constant but a random variable (akin to the lifespan of radioactive particles), then some of Adam's atoms would vanish before others. If the probability of any given atom vanishing stayed at zero (or stayed very low) for a period of time before rocketing up to be much higher, then time travelers who prevented their births would remain in existence for an interval before quickly fading away as their atoms collectively vanished over a brief stretch of time. LAW$_8$ can also explain cases of morphing objects. Consider again the example from *12 Monkeys*. If you scratch the earlier version of the watch, the later versions of those atoms will now never end up time travelling and so will blink out of existence, leaving a scratch in the time-travelling version of the watch [11].

However, LAW$_8$ has problems of its own, one superable and one insuperable.

The superable problem is one of libertarianism. LAW$_8$ features the worrying phrase 'causally responsible for'. To see why it is worrying, consider a deterministic scenario that involves no time travel: Bob shooting Bill dead at $t_{2023}$. Bill's death is caused by Bob's pulling the trigger. Whilst any court in the land would say that this cause was 'responsible for' Bill's death, metaphysics is less discriminating than jurisprudence—many other events are also responsible for Bill's death because, given determinism, at any time prior to $t_{2023}$, there is some collection of events that collectively cause Bill's death. That said, reconsider *The Adam Project*. When Adam$_{2050}$ and Adam$_{2022}$ arrive at $T_3$-$t_{2018}$, there is then *already* a collection of events that, by themselves, entail that Adam$_{2050}$ will not time travel back from 2050. The earliest cause for Adam not travelling in time is not some anthropocentrically interesting event (e.g., shooting the bad guy and foiling their evil plan at $t_{2018'}$) but is instead a massive collection of sub-atomic events at $t_{2018''}$, each seemingly uninteresting but which, collectively,

lead to that anthropocentrically interesting event. Call the conjunctive event consisting of those sub-atomic events '$e_{conjunction}$'. Since $e_{conjunction}$ takes place at $T_3$-$t_{2018}$ (rather than $T_3$-$t_{2018'}$), we have an inconsistency, at least if we assume—as seems reasonable, given what we see in *The Adam Project*—that there is only a short period of time between the prevention of $e_{travel}$ and Adam$_{2050}$/Adam$_{2022}$ vanishing from existence. Given that assumption, since $e_{conjunction}$ occurs at $T_3$-$t_{2018}$, LAW$_8$ would demand that Adam$_{2050}$ and Adam$_{2022}$ blink out of existence *before* $T_3$-$t_{2018'}$; whether they do or do not, we have trouble either way. If they do not blink out of existence, LAW$_8$ is false. If they do, then the absence of the Adams entails that (since the Adams will no longer exist at $T_3$-$t_{2018'}$) they no longer go on to prevent $e_{travel}$, and so $e_{travel}$ should occur, contrary to hypothesis. Thus, determinism is inconsistent with LAW$_8$.

We must abandon determinism. But we should not give up on it on the grounds that there are probabilistic quantum events. Instead, we must say that agents have libertarian free will. Given libertarian free will, there is some point in time when Adam makes an indeterministic decision that results in $e_{travel}$ being prevented; prior to that time, it is *false* that there is some conjunctive event like $e_{conjunction}$ that prevents $e_{travel}$. LAW$_8$ only starts the timer ticking on Adam$_{2050}$ and Adam$_{2022}$ remaining in existence when that free-willed action is enacted (i.e., when they bring about $e_{halt}$ at $t_{2018'}$). Thus, assuming that libertarian free will is closely possible (which is, admittedly, arguable), then this problem for LAW$_8$ can be solved. (And for a great example of free-willed decisions causing time-travelling objects to vanish/appear out of nowhere, listen to *The Hitchhiker's Guide to the Galaxy* [192].)

Turn next to the insuperable problem for LAW$_8$, which focuses on two features of it that I have suppressed by phrasing it as I did above. Having assumed that LAW$_8$ is reducible to the more fundamental laws of nature, we are assuming that the physical process that sends something back in time is likewise reducible. Call that process 'Z' and reparse LAW$_8$ so that it explicitly mentions $Z$:

LAW$_{8'}$ For all objects, $x$, which undergo process $Z$ at $T_j$-$t_k$, causing $x$ to travel to $T_{j+1}$-$t_{l<k}$: if, at a time later than $T_{j+1}$-$t_{l<k}$ (call it $T_{j+1}$-$t_{l<m<k}$) an event occurs which is causally responsible for preventing $x$ from undergoing $Z$ at $T_{j+1}$-$t_k$ then (at some time shortly after the free-willed action that prevents $x$ undergoing $Z$ at $T_{j+1}$-$t_k$), the version of $x$ that has undergone $Z$ vanishes from existence.

By parsing LAW$_8$ as LAW$_{8'}$, we highlight two things that the previous statement of it suppressed.

First: LAW$_{8'}$ recognises an extrinsic past-orientated property, i.e., 'having once undergone $Z$'. But, at the actual world, such extrinsic properties never feature in laws of nature—for instance, intrinsically identical electrons are always disposed to act the same way even if, say, one of them has previously passed through Shropshire. So, LAW$_{8'}$ has a metaphysical character different from the laws of the actual world (and, thus, is not closely possible).

Second: By talking about 'the version of $x$ that has undergone $Z$', LAW$_{8'}$ recognises numerical identity. Adam$_{2050}$ not travelling in time at $T_3$-$t_{2050}$ causes something to happen to Adam$_{2050}$ at $T_3$-$t_{2018''}$, i.e., his not time travelling causes something to happen to a thing *numerically identical to* him. Again, this is dissimilar to the actual laws of nature. The actual laws of nature say that if some thing with certain intrinsic properties is in certain circumstances, then a particular effect occurs; the laws of nature ignore whether that thing is or is not numerically identical to some other thing, caring only about its present qualitative intrinsic properties. The actual laws of nature are blind to diachronic numerical identity, whilst LAW$_{8'}$ needs the universe to have 20–20 vision on the matter. Again, the character of LAW$_{8'}$ is different from the character of the actual laws. Because of these two issues, LAW$_{8'}$ fails to be closely possible.

### 6.2. Morphing Non-Time Travelers

Having dealt with the morphing of time travelers, we move to the morphing of non-time travelers. In some fictions, the actions of a time traveler in the past cause non-time travelers in the present to cease to exist or otherwise morph. This happens in one of the

earliest time travel short stories, 'Ancestral Voices' [193], in which Emmet Pennypacker leaves 1935 and travels back in time 1483 years. Upon arrival, he kills someone, which causes fifty thousand descendants of that person to vanish from 1935. See Figure 4.

Similar examples can be found in other fictions [194–201]. And, again, we are not limited to people vanishing; not only do objects [202] or entire worlds vanish [203], but sometimes things change in other ways, e.g., writing on a wall appearing in the present as a time traveler scrawls on the wall back in the past [204], or a mark on a table appearing in the present 'simultaneous' with someone on the other end of an anti-telephone burning it by accident [180].

Another less common variant of the trope toys with the order of events. In the previous examples, the order of events is such that (at some given hypertime) a time traveler leaves in their time machine, and then a small amount of time passes before the activities of the departed time traveler cause things in the present moment to morph. This less common variant reorders those events by having the time traveler-to-be make a decision to act in their personal future, then the morphing takes place, *and only afterwards* does the time traveler go back to the past. For example, in *Meet the Robinsons* [53], Lewis, a child, travels to the future and has to defeat Doris, a robot that Lewis will create as an adult.

Lewis succeeds by forming the intention to never invent Doris, causing her to vanish. Afterwards, Lewis returns to the past, grows up, and purposefully fails to make Doris.

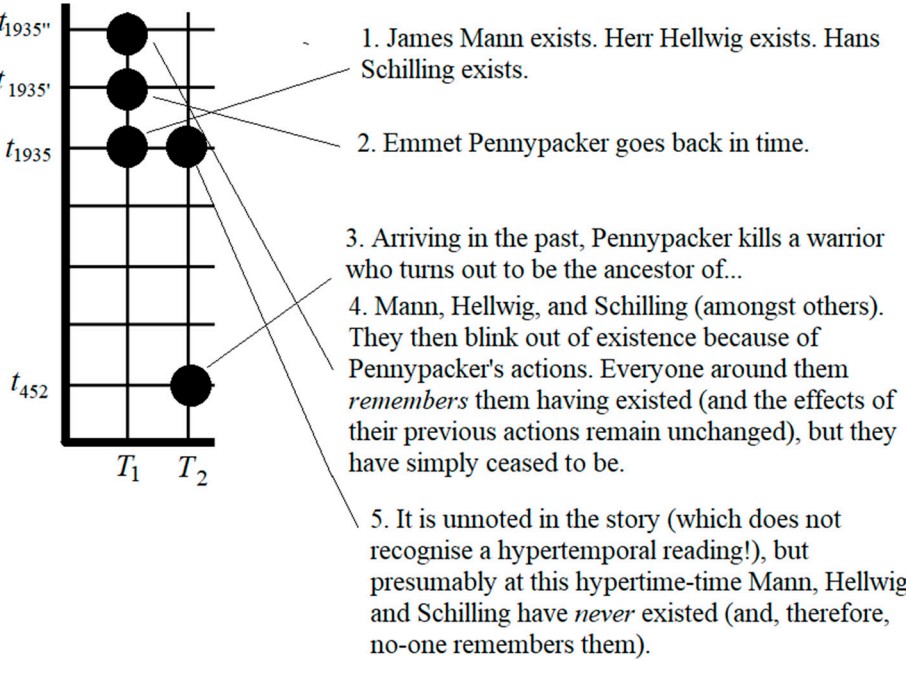

1. James Mann exists. Herr Hellwig exists. Hans Schilling exists.

2. Emmet Pennypacker goes back in time.

3. Arriving in the past, Pennypacker kills a warrior who turns out to be the ancestor of...

4. Mann, Hellwig, and Schilling (amongst others). They then blink out of existence because of Pennypacker's actions. Everyone around them *remembers* them having existed (and the effects of their previous actions remain unchanged), but they have simply ceased to be.

5. It is unnoted in the story (which does not recognise a hypertemporal reading!), but presumably at this hypertime-time Mann, Hellwig, and Schilling have *never* existed (and, therefore, no-one remembers them).

**Figure 4.** The narrative of 'Ancestral Voices' at a hypertemporal world.

For purpose of example, I will focus on a case where the morphing involves something appearing: *The Lake House* [205]. Kate sends messages to the past that are received by Alex, which then affect Alex's actions and which then change the future. At one stage, moved by Kate's letters about loving trees, Alex plants a tree outside some buildings. In the future, Kate is being rained upon outside the same building when, suddenly, the tree appears from nowhere as if it has always been there; Kate is surprised and shelters under it. Assuming the hypertemporal model, the narrative is thus: at $T_1$-$t_4$, Kate sends a letter to the past; at $T_2$-$t_1$, the letter is received by Alex; this causes him to plant a tree at $T_2$-$t_3$ (and no tree was planted at $T_1$-$t_3$); at $T_1$-$t_5$, Kate is caught in the rain (where there is still no tree); suddenly, a tree appears at $T_1$-$t_6$. See Figure 5.

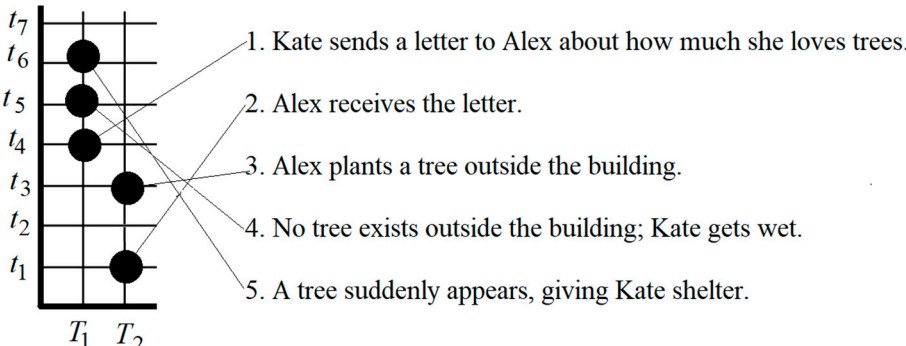

**Figure 5.** The narrative of *The Lake House* at a hypertemporal world.

Consider how the following law of nature deals with *The Lake House*:

LAW$_9$ If (i) at $T_j$-$t_k$, event $e$ occurs; (ii) $e$ causes $e'$ to occur in the past, at $T_{j+1}$-$t_{l<k}$; and (iii) $e''$ (which occurs at $T_{j+1}$-$t_{m>k}$) is caused by $e'$, then: $e''$ also occurs at the earlier hypertime, i.e., at $T_j$-$t_{m>k}$.

Given LAW$_9$, Kate's sending the letter (at $T_1$-$t_4$) causes Alex to plant a tree (at $T_2$-$t_3$), which causes there to be a fully grown tree at $T_2$-$t_6$ and thereby *also* causes there to be a fully grown tree at $T_1$-$t_6$. Alas, whilst LAW$_9$ goes some way to accounting for the narrative, it does not go all the way. Alex's planting of the tree also causes there to be a tree at $T_2$-$t_4$ and $T_2$-$t_5$; thus, LAW$_9$ dictates that there should then be a tree at $T_1$-$t_4$ and $T_1$-$t_5$, which is not what we see on-screen.

To get around this, we can conscript in ever more outré metaphysical theories. We need the combination of two such theories.

Theory one: An eccentric version of tensed theory. Like standard tensed theory, there is a privileged present moment moving forwards through time. The eccentricity is that when a privileged present moment interacts with a past moment, that past moment likewise becomes privileged; thus, in cases of time travel, there are two (or more!) privileged present moments travelling forwards through time [206] [207] (pp. 660–661) [208] (pp. 11–20). When Kate sends the letter to Alex, a second privileged moment comes into existence; call them NOW$_1$ and NOW$_2$, respectively. NOW$_1$ and NOW$_2$ trundle forwards through time, NOW$_1$ forever remaining at $T_1$ and NOW$_2$ forever remaining at $T_2$, with NOW$_2$ always being earlier in time than NOW$_1$; see Figure 6.

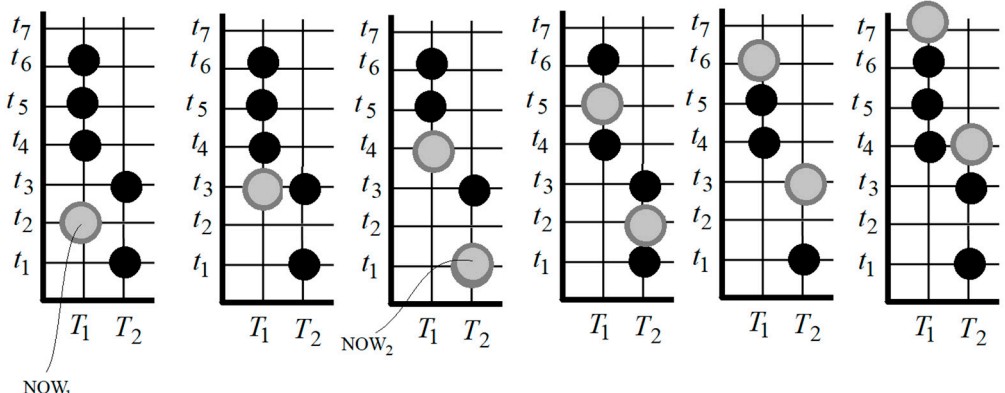

**Figure 6.** The progress of NOW$_1$ and NOW$_2$ as time passes. Black dots represent events (as per Figure 5); the grey circles represent NOW$_1$ and NOW$_2$.

Theory two: A specific version of libertarian free will: Geachianism [165,209,210]. Geachians believe that there are contingent truths about the future but that these truths change as agents make free-willed libertarian choices. For instance, assuming it is presently

2023, it might be presently true that Malcolm, a smoker, will die in 2040 from cancer. When, in 2024, Malcolm later makes the free-willed decision to quit cigarettes, it then becomes *false* that he dies in 2040 from cancer (and it becomes true instead that, say, in 2040, he is running a marathon). (Note that the actions of agents have such a radical effect on the future that Geachianism entails that agents play a fundamental role within reality's structure. We might think that this alone means that morphing is not closely possible, but I will set that worry aside for much the same reason I set aside Section 4.2.'s commitment to substance dualism.)

Combining these theories, we can move towards a solution to the problem at hand. Libertarian choices necessarily take place in the privileged present. So, at $T_2$, prior to Alex making the decision to plant the tree, there is no tree at later times. When $NOW_2$ reaches $T_2$-$t_3$, Alex freely plants the tree. Given Geachianism, it is now true that (at $T_2$) at $t_4$, $t_5$, $t_6$, etc., a tree exists there. Since this happens when $NOW_2$ is at $T_2$-$t_3$, the hypertime-times that now include a tree (i.e., $T_2$-$t_4$, $T_2$-$t_5$, $T_2$-$t_6$ etc.) are not privileged. But here's the rub: Whilst those hypertime-times fail to be metaphysically privileged, one of them is nevertheless special in some other way since it is at the same time—though not the same hypertime!—as $NOW_1$; I shall say that such a hypertime-time is 'paraprivileged'. So, assuming that $NOW_1$ is at $T_1$-$t_6$ when Alex plants the tree at $NOW_2$, then $T_2$-$t_6$ is paraprivileged; see Figure 7. Just as privileged hypertime-times make for joints in nature, so too do paraprivileged times; so, consider:

LAW$_{10}$ If (i) $NOW_n$ is a privileged present currently located at $T_{j+1}$-$t_k$; (ii) (at $NOW_n$, i.e., at $T_{j+1}$-$t_k$) an agent makes a free-willed libertarian choice, making it the case that event *e* occurs at some future time, $T_{j+1}$-$t_{l>k}$; and (iii) $T_{j+1}$-$t_{l>k}$ is paraprivileged, then: *e* also occurs at $T_j$-$t_{l>k}$.

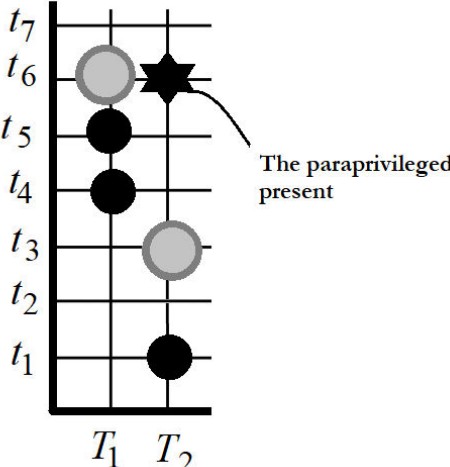

**Figure 7.** The paraprivileged present (given a specific arrangement of $NOW_1$ and $NOW_2$). Black dots represent events (as per Figure 5); the grey circles represent $NOW_1$ and $NOW_2$.

LAW$_{10}$ gets us closer to explaining the narrative of *The Lake House*. Consider one of the events that takes place at $T_2$-$t_6$: the event of the grown tree growing that little bit more. Since $T_2$-$t_6$ is paraprivileged, LAW$_{10}$ entails that this event 'translates over' to $T_1$-$t_6$ and a fully grown tree springs into existence at $T_1$-$t_6$ (for, if every cell in the tree 'grows that little bit more' at $T_1$-$t_6$, then the entire tree must exist at $T_1$-$t_6$, and, thus, an entire tree must pop into existence). Moreover, whilst the tree exists and grows at earlier times at $T_2$, e.g., at $T_2$-$t_4$ and $T_2$-$t_5$, since those hypertime-times are *not* paraprivileged, then that event does *not* occur at those hypertime-times (and so there is not a tree at $T_2$-$t_4$ and $T_2$-$t_5$). So, we have overcome the problem we were faced with.

But, even having deployed this heavy-duty metaphysical apparatus, there is still a problem with $LAW_{10}$ accounting for the narrative of *The Lake House*. When Alex plants the tree at $T_2$-$t_3$, he not only makes the future different regarding the tree, but he makes all sorts of other things different. For instance, he makes Kate different. The Kate at $T_2$-$t_6$ remembers the tree as always having been there; similarly, that Kate never got wet in the rain and shows no surprise at there being a tree, etc. Given $LAW_{10}$, when Alex plants the tree, he should not only make a tree appear out of nowhere at $T_1$-$t_6$, but he should also change Kate—she should suddenly act as if the tree was always there and should suddenly be dry rather than wet, and she should show no signs of surprise, etc. Since this is not what we see on-screen, $LAW_{10}$ fails to capture the narrative.

What we need is a law whereby only *certain* events occurring at the paraprivileged time are forced to occur at the corresponding privileged present at the hyperearlier hypertime. But systematically distinguishing between those events that are forced to occur and those that are not is a big challenge! We cannot even mirror the move of Section 5.3 and say that it is only those events that are of 'anthropocentric interest' that are forced to occur since a tree being in a given place or not seems about as roughly as anthropocentrically interesting as Kate being dry as opposed to wet.

But perhaps there is a way. Having already included libertarian free will, we could again put it to work. Alex is the sole agent causally responsible for the existence of the tree at $T_2$-$t_6$—whilst other things are responsible (e.g., rain falling from the sky and the photosynthesis of the tree), they are not agents. Whereas, when Kate fails to be surprised at $T_2$-$t_6$, that is a result of her actions; her libertarian free will plays a role in that. So, consider:

$LAW_{11}$ If (i) $NOW_n$ is a privileged present currently located at $T_{j+1}$-$t_k$; (ii) (at $NOW_n$, i.e., at $T_{j+1}$-$t_k$) an agent makes a free-willed libertarian choice, which makes it the case that event $e$ occurs at some future time, $T_{j+1}$-$t_{l>k}$; (iii) that agent is the sole agent playing a causal role in bringing about $e$; and (iv) $T_{j+1}$-$t_{l>k}$ is paraprivileged, then: $e$ also thereby occurs at $T_j$-$t_{l>k}$.

$LAW_{11}$ avoids the problem and, I suggest, allows for the close possibility of *The Lake House*. However, given the amount of outré metaphysical machinery required to make this solution work—and worries about whether this move can be extended to other fictions featuring morphing non-time travelers—the extent to which this is a victory, I leave to the reader to decide [12].

## 7. Other Tropes?

The worries that have been discussed for the above three tropes can be extended to undermine the close possibility of other time travel tropes.

Consider the trope of displacement, whereby when someone time travels, they either cause their earlier self to vanish (or to switch places with them) [211] or cause someone else to vanish (or switch places with them) [120,212]. In the case of the former, we run into the problem from Section 6.1, whereby numerical identity ends up being enshrined in a law of nature. In the case of the latter, we run into problems of ontological inegalitarianism and reducibility: why is it always another *person* who is displaced when a *person* travels in time? What if a laptop travels back in time—would it displace other laptops, and, if so, does the sortal term 'laptops' thereby feature in the laws of physics? And if a laptop was taken back to the 1960s, would it displace a more primitive computer? Would it displace an abacus when taken back to Ancient Greece? These questions have nothing but embarrassing answers. Whilst a 'sortal neutral' version of displacement may be closely possible—e.g., requiring only that sending a time traveler into the past displaces an equal amount of *mass* back into the future [213]—most cases of this trope will not be [13]. Consider the trope of life/death exchanges, whereby going back in time and saving a life requires someone else to die [195,216]. Presumably, there is no corresponding 'exchange law' for sortals other than 'living humans'. For instance, if I make a cup of tea in the past, does that mean that some other cup of tea must now go unmade? Presumably not—it is only the bringing into existence of, or continuing the existence of, living humans that must be balanced out by

the ceasing to exist of some other living human. In that case, the laws will not be closely possible because the laws are not ontologically egalitarian, are not reducible, etc.

Consider the trope of floating clock syndrome, whereby when one time travels, clocks, calendars, or other time-measuring devices appear [69,217–220]. Presumably, this is a contingent relationship—i.e., it is not clocks and calendars which would be called into existence if, instead of humans evolving, aliens had evolved and made 'dlocks' and 'dalendars' (and other alternative time-measuring devices). In that case, when the aliens time travelled, then dlocks and dalendars would pop into existence. But then the laws of nature would not be invariant, failing to be the same wherever/whenever you are. Again, they would not be closely possible.

In short, the same sorts of arguments used against the three tropes discussed in Sections 4–6 can be deployed against other tropes as well.

For some such tropes, the arguments will find purchase in some contexts but not others. Consider the trope whereby organic things, such as people, can time travel, but synthetic things, such as clothing or weapons, cannot [221,222] (p. 46) (see also [153,223]). Is that closely possible? Whilst the actual laws do not respect an organic/synthetic distinction, we may think that—in at least some contexts—fictional laws that did respect such a difference are nevertheless of the same *character* as the actual laws. Equally, there are contexts in which we would say the opposite, e.g., if one is aiming for 'hard science fiction', doubtlessly, such a law would rile. This trope's close possibility is, thus, context-dependent.

## 8. Broader Possibility

Having considered close possibility, I now turn to considering the broader modalities, e.g., logical and metaphysical possibility. This section argues that whether or not the tropes are more broadly possible depends upon what metaphysics of the laws of nature one thinks is broadly possible.

### 8.1. Humean Laws of Nature

Across spacetime, there is a distribution of perfectly natural qualities; for instance, at one spacetime point, *Charge* and *Spin-up* are jointly instantiated, whilst at another, no natural properties are instantiated (and so on for all points). Humeans say that the laws of nature metaphysically depend upon that distribution; non-Humeans say the opposite. Humeans say that it is a law of nature that electrons have a mass of 0.51 MeV because all electrons have a mass of 0.51 MeV; non-Humeans, on the other hand, say that electrons have the mass that they do because of the laws of nature. This sub-section considers what Humeans will say about the logical and metaphysical possibility of the tropes.

Humeans usually accept a liberal principle of recombination. There is a debate about how best to cash out such a recombinatorial principle [224–227]. This paper takes it to be the claim that, for any fundamentalia, it is possible for there to be any number of duplicates of those fundamentalia (where zero is a number), standing to one another in any perfectly natural relation (where 'possible' picks out some species of broad possibility, which may vary to yield a variety of different strengths of recombinatorial principle). Given such a principle, there are worlds that are intrinsic qualitative duplicates of the fictional worlds we have thus far considered. For instance, given recombination, there is a world where a young Nova-esque girl gets out of a craft claiming she is from the future, followed later by an older Nova-esque woman getting into such a craft and vanishing (indeed, recombination even allows for the hypertemporal readings of the various fictional narratives [18] (pp. 86–88)).

But, whilst it is broadly possible for there to be intrinsic duplicates of the fictional worlds that we have considered, it does not follow that the fictional worlds are, therefore, broadly possible. For the fictional world to be broadly possible, propositions about things other than intrinsic qualitative concerns must also be true at it, namely the explanatory propositions. For instance, the possible world containing Nova-esque things is only identical to the fictional world of *Captain Nova* if the event of the older Nova-esque woman activating the craft *explains* the arrival of the younger Nova-esque woman back in the past.

Given Humeanism, those explanations will hold (and, thus, the fictional world will be broadly possible). Imagine that, at that world, thousands of time travelers return back in time, each deaging as they head back into the past. Just as the law concerning electron mass is metaphysically determined by the masses of the zillions of electrons, were every time traveler to deage, then Humeanism would have it that it is thereby a law of nature that time travelers deage. *Captain Nova* may not be closely possible, but the Humean will say that it is broadly possible (specifically: if both Humeanism and recombination are logically possible, so too is *Captain Nova*; if both Humeanism and recombination are metaphysically possible, so too is *Captain Nova*; and so on).

*8.2. Non-Humean Laws of Nature*

There are many non-Humean alternatives to Humeanism. To simplify matters, I focus only on one such theory: the laws of nature are metaphysically determined by natural necessitation relations holding between universals. For instance, the law of electron mass holds because two universals (i.e., *Electron* and *0.51 MeV*) stand in a natural necessitation relation [228–230].

Non-Humeans might be less friendly to the recombinatorial principle, which is more regularly associated with Humeanism. If so, they might then deny that the fictional worlds are broadly possible on the grounds that intrinsic duplicates of those worlds are not possible (e.g., there is not even a world containing the Nova-esque things, never mind a world where those Nova-esque things are numerically identical and at which the correct explanations hold). But, for the sake of argument, let us assume that the non-Humean finds a relevant recombinatorial principle to be attractive, allowing us to focus our attention on whether or not the relevant laws of nature hold at those worlds. Given non-Humeanism, that amounts to whether or not there is a world where (i) the appropriate universals exist and (ii) they stand in the relevant natural necessitation relation. For example purposes, we shall stick to considering *Captain Nova*. Above, I said that the following needed to be a law of nature at the fictional world of *Captain Nova*:

LAW$_2$ Any organism that travels back in time *n* years regresses in age *n* years.

Our non-Humean, therefore, (i) needs for there to be two (quantitative) universals, *Travels Back in Time __ years* and *Is an independent organism of __ years of age*, such that (ii) those universals are related by a natural necessitation relation.

Consider (i). As noted in Section 6.1, travelling back in time could be a natural process recognised by the laws of nature; thus, I do not think it is problematic to believe that *Travels Back in Time __ years* is a metaphysically or logically possible universal. However, whether or not *Is an independent organism of __ years of age* is broadly possible largely depends upon what one says more generally about the possible existence of structured universals. Consider predicates like:

　　__ is a rock star
　　__ once had a teenage crush on Carrie Fisher
　　__ is made of paints sourced from South America and applied by the artist Jackson Pollock

Those who are liberal about possible universals will say that those predicates could correspond to a universal. More conservative non-Humeans will say that they could not. So, liberals will allow for the possibility of a fictional world with the relevant universal (i.e., *Is an independent organism of __ years of age*), whilst conservatives will demur (and, therefore, will demur over the broader possibility of *Captain Nova*).

Liberal non-Humeans who allow (i) might, nevertheless, go on to deny (ii). To start with, consider a *bad* reason for thinking that liberals will believe (ii): (ii) is true because recombination is true. The thought would be that because natural necessitation is a perfectly natural relation, recombination entails that there is some world where those two universals stand in the relevant connection of natural necessitation. But recall that recombination deals only in fundamentalia; it does not say that derivatives can be freely combined (else there would be a world at which *Electron* and *Charge* naturally necessitated one another,

as did *Electron* and *Not Charged*, which would be contradictory). Since neither *Travels Back in Time __ years* nor *Is an independent organism of __ years of age* are candidates for being fundamental, there is no particular reason for the non-Humean to believe that the former can naturally necessitate the latter.

Indeed, not only is there no reason to believe it, but there is also a reason to believe that the universals *cannot* stand in natural necessitation relations. Derivative universals may well stand in natural necessitation relations (e.g., *Electron* and *Charge* stand in such relations), but—plausibly—they only do so because their constituent parts themselves stand in some appropriate network of natural necessitation relations (i.e., the necessitation between derivative universals supervenes on the natural necessitation relations between their constituent parts). If true, that seems to be synonymous with saying that the laws of nature that involve derivative universals are always *reducible* laws. Since the laws needed for the possibility of the above time travel tropes are generally *irreducible*, the non-Humean then has good reason to think that they are not broadly possible.

In short, whether or not these time travel tropes are broadly possible will depend upon your chosen metaphysics of the laws of nature. If you are a Humean, you are likely going to be attracted to believing that the tropes are broadly possible. If you are a non-Humean, things are trickier, although I have suggested that there is good reason to believe that they are not broadly possible.

### 9. What Might We Mean by 'Is this Fiction Possible?'

This paper has argued that certain time travel tropes are not closely possible but may—depending on your favored metaphysics of laws—be more broadly speaking possible. (This revises my previous position on these matters [18] (pp. 111–112).)

Further to this, I hope to have given a better perspective about what one is *doing* when one asks whether a time travel fiction is possible or not. The early debates about the philosophy of time travel, from the 1960s and 1970s, explicitly focused on the broader modalities [1,12,15,24,142,145,231–237], but most people who complain about whether a time travel story is possible or not (such as non-philosophers) will likely have something else in mind. This paper makes it seem likely that it is 'close possibility' that such people have in mind. (Although, there is another alternative, which I have not explored as much in this paper: When one says that a time travel trope is impossible, one might merely be saying that it is not broadly possible that the trope has a purely logical explanation, i.e., nothing about *time travel alone* explains the time travel trope, even though some contingent physical law may nevertheless shoulder that explanatory burden. I leave the merits of that alternative for the reader to think about.)

**Funding:** This research received no external funding.

**Acknowledgments:** Thanks to the two anonymous referees for this journal, whose comments have been incorporated into this paper. Thanks also to Iain Law, Alex Northover, Henry Taylor, Martin Pickup, and Keith Jones. A special thanks to Ema Sullivan-Bissett, especially for her advice on function, which greatly influenced Section 4.2.

**Conflicts of Interest:** The author declares no conflict of interest.

## Notes

1   Figure 1 assumes that physically possible worlds are all metaphysically possible, contrary to arguments I have made elsewhere [18] (pp. 128–143). Since that issue is irrelevant to this paper, I ignore it entirely.

2   Experience teaches me that introducing a time travel trope via examples is best accompanied by examples of what *does not* count as being an example of the trope. To that end, consider cases where a time traveler deages by 'mentally time travelling' into a past body that happens to be younger. This is common in 'time loop fiction' like *Groundhog Day* [81], as well as its precursors [82,83] and imitators [84–88]. Likewise, time travelers may come to inhabit either their earlier selves [89–92] or some other, younger person [93–96]. These cases are *not* examples of the 'deaging' trope—I have in mind only cases where one's own *physical body* is caused to change age via the act of time travel (or via some time travel-related device or process).

³ Likewise, if one regresses to a very young age [67], the skull will not fit an adult-sized brain. Such time travelers would suffer incredibly messy deaths.

⁴ Further to footnote 2, we can finesse what does and does not count as an example of the deaging trope. In some fictions, time is sped up, frozen, or reversed in a given spatial region [100,101]. Such cases—which, again, are not dissimilar to the real-world phenomenon of time dilation—are *not* cases of the deaging/aging trope that I have in mind during the discussion in the main text.

⁵ More exotically: If an organism previously had parts *of a type* that it does not later have, what happens then? For instance, if the Fictionoid people of Mythtopia have juvenile cardiovascular systems made of exotic matter, which they shed when they undergo metamorphogenesis in becoming adults, what happens when mature Fictionoids time travel? Do they arrive in the past *sans* cardiovascular systems and then immediately die? If so, that seems weird; if not, that is a problem for $LAW_4$. (Of course, since we never see a 'Fictionoid' on-screen, one might reasonably bite the bullet and just accept the weirdness that we never witness.)

⁶ We can also consider non-time travel cases. For instance, in *Dogma* [116], attempts to make false one of God's assertions threaten the entirety of reality.

⁷ Again, as per footnote 2, consider what does *not* fall under this trope. In many fictions, the past is Ludovician (*q.v.*) and attempts to change the past are thwarted. Whilst most fictions have it that the thwartings are relatively innocuous, some fictions have it that a terrible disaster plays the thwarting role [129–131]; indeed, elsewhere, I have argued that such threats are an *actual* danger, not merely a fictional one [18]. Such 'Ludovician thwartings' are *not* an example of the 'disaster trope' discussed in the main text. The disasters of the main text are dissimilar from Ludovician thwartings because their cause is an actually occurring paradox rather than the mere threat of an actually occurring paradox.

⁸ This definition, with its use of 'immediately hyperprior', requires hypertime to have a discrete, not a continuous, ordering. Note, then, that elsewhere [4] (p. 19) I have already argued that, if hypertime is to be deployed to help with time travel, hypertime must have a discrete ordering.

⁹ This assumption of determinism is probably dispensable. Ultimately, all I need is that any given hypertime evolves the same as the immediately hyperprior hypertime, except when time travel occurs. Elsewhere, I have discussed how this could be the case without determinism being true [4] (pp. 3–8).

¹⁰ Again, consider examples of what this trope does *not* involve. In some fictions, time travelers prevent themselves from being born [182,187–189], thus—in some sense—causing themselves to 'cease to be'. But, as long as the *time traveler* remains in existence, this is not an example of the trope under consideration in the main text because such time travelers have only caused versions of themselves from *hyperlater hypertimes* to not exist, rather than (as in the main text) the time traveler themselves.

¹¹ One upshot would be that since composite objects change their parts, a composite object can immunise itself from fading away. If Adam₂₀₅₀ and Adam₂₀₂₂ held off from causing $e_{halt}$ for long enough, then all of the atoms in their body would be replaced by non-time-travelling equivalents. When $e_{halt}$ then occurs, the atoms that *previously* composed them will go on to vanish, but that would make no difference to Adam₂₀₅₀ and Adam₂₀₂₂, who would remain in existence.

¹² We would also need a clear distinction between positive and negative causation since every agent who walks past the tree causes the tree to keep growing insofar as they *do not* choose to chop it down with an axe. Plausibly, though, Alex is the sole agent who *positively* (partially) causes it to be there.

¹³ As noted in footnote 2, some fictions allow for consciousness alone to time travel. In at least one short story [214], there is a ban on someone's consciousness inhabiting a body at the same time as its earlier self. Interestingly, given certain assumptions, this sort of fiction is closely possible! It is common to believe that, by metaphysical necessity, two disjoint things cannot partially occupy the same point at the same time. Given substance dualism, concrete souls would occupy 'merely temporal' points [215]; given the ban on co-location, for each soul, there would then have to be a distinct collection of 'merely' temporal points, with each collection standing in one-to-one correspondence with the instants of time, forming individual 'soul streams' for Cartesian souls to occupy. If all of this were true, we could explain the impossibility of a consciousness returning to a time it previously existed at; for it to return, it would have to occupy a point previously occupied by some soul (namely, itself!) and, since that is impossible, it will transpire that no entity can send its consciousness back to a time that it already exists at.

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
