# Peer review of "The Close Possibility of Time Travel"

_philosophies, doi:10.3390/philosophies8060118_

Round 1

Reviewer 1 Report

Comments and Suggestions for Authors

Some relatively minor comments:

Line 9: “and arguing”—“and argue”?

Line 76.5.  Figure 1.  Are the ‘close possibility’ and ‘physical possibility’ labels reversed given what author says at lines 70 and 72?

Line 210.  “Instead, substance dualism must be true.”  Must be true? Certainly, the brain cannot be identical with the 12-year old brain and her mind remain the same, but physicalists allow for multiple realizability, so some physical structure (a 12-year old sized brain with more neural connections) might still realize the 37-year old mind.  Of course, then the issue becomes why the brain does not regress, but rather becomes more neurally compact say, while everything else in her body simply regresses.

Line 324. “assume an arrangement which they should have, conditional on …”. I’m not getting why this description of what is going on in the Nova case is true.  Granted the 37-year-olds parts cannot assume the position they had 25 years ago and get the desired result.  What is wanted is something like: her 37-year old parts, as much as possible, arrange themselves as her 12-year olds parts were arranged. (Perhaps some sort of switching of location of Nova’s fundamental parts would be required.)  But appealing to how her parts were in fact arranged, does not appeal to blueprint facts about how her parts should be arranged.  Of course, this alternative way to modify Law3 will likely fail to be acceptably similar to current laws.

Line 479: Get rid of the gap between Law5 and “If”

Line 549: Get rid of gap between Law6 and “If”

Line 751: Law 9 (also applies to Law 10).  So both of these laws appeal to ‘agents’.  Will this require yet more theoretical baggage to make agents either reducible or a fundamental kind?  [Not clear that Geachianism by itself entails ‘agents’ are a natural or fundamental sortal.]

Author Response

Minor issues fixed. I've also added some extra material to respond to comments.

>>>>Line 210.  “Instead, substance dualism must be true.”  Must be true? Certainly, the brain cannot be identical with the 12-year old brain and her mind remain the same, but physicalists allow for multiple realizability, so some physical structure (a 12-year old sized brain with more neural connections) might still realize the 37-year old mind.  Of course, then the issue becomes why the brain does not regress, but rather becomes more neurally compact say, while everything else in her body simply regresses.

For response, see the very bottom of p. 5 and top of p. 6 of the revised manuscript.

>>>>>Line 324. “assume an arrangement which they should have, conditional on …”. I’m not getting why this description of what is going on in the Nova case is true.  Granted the 37-year-olds parts cannot assume the position they had 25 years ago and get the desired result.  What is wanted is something like: her 37-year old parts, as much as possible, arrange themselves as her 12-year olds parts were arranged. (Perhaps some sort of switching of location of Nova’s fundamental parts would be required.)  But appealing to how her parts were in fact arranged, does not appeal to blueprint facts about how her parts should be arranged.  Of course, this alternative way to modify Law3 will likely fail to be acceptably similar to current laws.

For this, see p. 8 of the revised manuscript.

>>>>>Line 751: Law 9 (also applies to Law 10).  So both of these laws appeal to ‘agents’.  Will this require yet more theoretical baggage to make agents either reducible or a fundamental kind?  [Not clear that Geachianism by itself entails ‘agents’ are a natural or fundamental sortal.]

Added clarificatory comment. See page 18 of the revised manuscript.

Reviewer 2 Report

Comments and Suggestions for Authors

I recommend this paper for publication. I must confess that I was initially skeptical that a thorough discussion of various time travel tropes and "close possibility" would prove sufficiently substantive, but the author has done an excellent job arguing otherwise. I found the various discussions both insightful and enjoyable. I am certain that it will attract attention both from specialists in the field as well as more general time travel enthusiasts who are interested in some of the most enduring (and puzzling) time travel tropes in science fiction. I don't believe the paper needs to be revised (other than the references, as mentioned by the author) before publication. Thank you for the enjoyable read!

Author Response

Thanks! (References now all fixed and finalised)